# Enrichment of Mango Fruit Leathers with Natal Plum (*Carissa macrocarpa*) Improves Their Phytochemical Content and Antioxidant Properties

**DOI:** 10.3390/foods9040431

**Published:** 2020-04-04

**Authors:** Tshudufhadzo Mphaphuli, Vimbainashe E. Manhivi, Retha Slabbert, Yasmina Sultanbawa, Dharini Sivakumar

**Affiliations:** 1Department of Horticulture, Tshwane University of Technology, Pretoria West 0001, South Africa; chudufhadzo@gmail.com (T.M.); slabbertmm@tut.ac.za (R.S.); 2Phytochemical Food Network Group, Department of Crop Sciences. Tshwane University of Technology, Pretoria West 0001, South Africa; vimbainashed@gmail.com; 3Australian Research Council Industrial Transformation Training Centre for Uniquely Australian Foods, Queensland Alliance for Agriculture and Food Innovation, Center for Food Science and Nutrition, The University of Queensland, St Lucia, QLD 4069, Australia; y.sultanbawa@uq.edu.au

**Keywords:** cyanidin-3-O-glucoside, cyanidin-3-O-β-sambubioside, epicatechin, agro processing, phenols, food security, proximate composition

## Abstract

Natal plum fruit (*Carissa macrocarpa*) is indigenous to South Africa and a rich source of cyanidin derivatives. Indigenous fruits play a major role in food diversification and sustaining food security in the Southern African region. Agro-processing of indigenous are practiced adopted by the rural African communities in order to reduce the postharvest wastage of fruit commodities. In the current study, Natal plum was added to mango pulp at different ratios (mango and Natal plum (5:1, 3:1, 2:1)) to develop a healthy-functional snack (fruit leather). The effects of added Natal plum on the availability of antioxidant constituents and in vitro antioxidant properties of a mango-based fruit leather were evaluated by comparing with mango fruit leather. Fruit leather containing mango and Natal plum (2:1) retained the highest content of cyanidin-3-O-glucoside chloride, cyanidin- 3-O-β-sambubioside, epicatechin, apigenin, kaempferol, luteolin, quercetin-3-O-rhamnosyl glucoside, catechin, quinic, and chlorogenic acids, and in vitro antioxidant activity. Proximate analysis showed that 100 g of fruit leather (2:1) contained 63.51 g carbohydrate, 40.85 g total sugar, 0.36 g fat, and 269.88 cal. Therefore, enrichment of mango fruit leather with Natal plum (2:1) increases its phytochemical content and dietary phytochemical intake, especially for school children and adolescents.

## 1. Introduction

Natal plum (*Carissa macrocarpa*) fruit is indigenous to South Africa, has an attractive red colour (Figure 1), and is a rich source of cyanidin derivatives (cyanidin-3-O-glucoside, cyanidin-3-O-β-sambubioside, and cyanidin-3-O-pyranoside) [1]. These cyanidin derivatives are associated with health benefits such as anti-inflammatory, antiviral, anti-proliferation, and anti-carcinogenic effects [2]. Natal plum is also rich in vitamin C, calcium, magnesium, and phosphorus [3]. Urbanization, change in food habits, and sustainable food systems contribute more to hidden hunger, and it is more prevalent among the urban population [4]. Therefore, diet diversification with traditional underutilized fruits can be one way to tackle hidden hunger [4]. Traditional underutilized fruits are readily available during different seasons; they are easily harvestable and cost much lower price than commercial fruits such as citrus or avocado. Therefore, Natal plum can be included in food diversification and traditionally rural communities make jams and jellies from this fruit. Thus, this fruit has the potential of improving rural industry and well-being of the communities. Based on the nutritional facts, Natal plum can be introduced in nutrition intervention programmes to sustain food and nutrition security.

Fruit leather is a dried-fruit or dehydrated fruit, high in fibre and carbohydrates, whilst low in fats. Fruit leathers are chewy, have a pleasant flavor, and are consumed as a sweet snack [5]. Due to its appealing nature, dried fruit leathers are another practical way to increase the consumption of solid fruit, especially for children and youth, or for communities affected by natural disasters or in war regions where food is in secure [5].

Since fruit leathers are concentrated, with a higher nutrient density compared to fresh fruit due to dehydration, this makes them a healthier and convenient alternative snack compared to candies and confections [6]. In addition, fruit leathers contain fewer calories per serving [7] and greater nutritional value in terms of antioxidants and minerals due to the dehydration process concentrating the nutrients [8]; therefore, they are suitably healthy food for heath food markets. The lower moisture content of fruit leathers also reduces the microbial infestation during storage and transportation.

Mangoes are widely produced in Venda Limpopo province at the village level. Each household has 3 to 4 mango trees in their home garden. The community cooperatives do not have infrastructure and a cold chain facility to store the mango fruits. Therefore, postharvest losses of mangoes during the mango season due to surplus supply are very high. Thus, using fresh mango pulp for fruit leather preparation is one way of preserving mangoes during off-season. It has been standard practice to add pectin, sugar, and sodium metabisulphite to fruit leathers; however, shelf-stable and acceptable mango (*Mangifera indica*) leather has been produced without any sugar and preservatives [9]. This makes mangoes ideal for making a leather, which suits consumer demands for natural and additive-free snacks. Apart from the aforementioned, mango fruit is well liked by consumers due to its attractive colour and flavour; it is also regarded as a good source of dietary antioxidants, such as ascorbic acid and carotenoids [10]. Mango pulp is rich in magnesium, potassium, and other essential nutrients [11]. Consumption of mango pulp effectively favorably affected postprandial glucose and insulin responses in individuals with Type 2 diabetes. The dietary source of carotenoids, flavonoids, and various phenolic acids and mangiferin present in mango pulp was proven to reduce blood glucose levels by inhibiting glucose absorption from the intestine, demonstrating antidiabetic properties [12]. Since mango pulp also contains some pectin, this makes it best suited for fruit leather application [13].

Inclusion of maqui berry extract to apple or quince leathers improves the functional properties and colour of the leathers from light brown to dark purple [14]. The enrichment of apple leathers with raspberry and blackcurrant improves their phenolic composition and antioxidant activity [15]. Adding Natal plum to mango pulp could increase the antioxidant properties of the mango–Natal plum fruit leather. In light of the aforementioned, the objective of the current study is to evaluate the incorporation of a Natal plum (berry) in different proportions to mango-based leather for the improvement of antioxidant constituents and to study in vitro antioxidant activity. Proximate analysis (total carbohydrate, sugar, fat, and energy) was performed to obtain information on the best Natal plum and mango pulp ratio that offers the highest antioxidant properties.

## 2. Materials and Methods

### 2.1. Sample Collection and Preparation

Natal plum (*Carissa macrocarpa*) fruits were harvested fresh from Tshwane University of Technology (TUT) main campus, Orion Residence, Pretoria, Gauteng Province. Mango fruits (*Mangifera indica* cv. Tommy Atkins) were selectively hand-harvested for this study, from home gardens in Venda (Vuwani), and transported to TUT at 10 °C within 6 h after harvest. Mango (brix° 20) and Natal plum (brix° 10) fruits were sanitised using free active chlorine solution (in a 50 mg L^−1^), as recommended by Azeredo et al. [10]; thereafter, the mangoes were peeled, de-seeded, and cut into small cubes of 2 cm^3^ each. Natal plums were cut into two halves and both types of fruit were weighed to determine the proper ratio required for the fruit leather preparation. In this study, the fruits were not cooked in order to prevent the loss of phytochemicals. Cut mango fruit was mixed with Natal plum fruit in three different ratios—5:1 Mango (M) Natal plum (N) MN5, 3:1 MN3, 2:1 MN2, and afterwards pureed using a food processor (Bosch, MCM3301BGB, Milton Keynes, UK) for 5 min. The resulting mixture was layered (9–10 mm thickness) on a non-stick sheet and dehydrated using a drying oven (Universal oven Memmert GmbH, Buechenbach, Germany) at 60 °C for 4 h until a moisture content of 15% was reached. Mango (M1) and Natal plum (N1) leathers were also made. The resulting fruit leather was cooled, cut into pieces (6 cm^2^), rolled up, and packed in BOPP (water vapor transmission rate of 4 × 10^−3^ kg/m^2^/d at 90% RH, 38 °C and an OTR of 2.5 L/m^2^/d atmosphere at 25 °C to determine fruit leather colour properties. A set of 10 replicates of each fruit leather were snap-frozen in liquid nitrogen and stored at −80 °C for antioxidant and phytochemical analysis.

### 2.2. Fruit Leather Colour Properties

Colour properties of mango, Natal plum, and the mango-Nata plum mixed leathers were objectively measured at three points of the fruit leathers (two each on the opposite side and bottom) using a colour meter, Minolta CR-400 chromameter (Minolta, Osaka, Japan) [16]. A standard white tile was used to calibrate the chromameter. According to the colour system used, *L** measures lightness from black to white (0–100); *a** indicates red (+) to green (–), while *b** measures yellow (+) to blue (–). The intensity of the red colour is represented by positive *a** values, whereas the intensity of the yellow colour is represented by positive b* values. Furthermore, the *h˚* values, known as a colour wheel, consist of red-purple angled at 0°, yellow at 90°, bluish-green at 180°, and blue colour at 270°. The total colour difference ΔE = [(ΔL*) ^2^ + (Δa*) ^2^ + (Δb*) ^2^] ^½^ was calculated according to which ∆L = colour difference is calculated as the sample *L** value minus standard, ∆*a** = colour difference is calculated as the sample *a** value minus standard, ∆*b** = Colour difference is calculated as the sample *b** value minus standard [16].

### 2.3. Total Phenols

Total phenolics were determined using the Folin-Ciocalteu reagent [1,17]. Snap frozen fruit leather of 0.1 g was homogenised in 2 mL of 80% methanol containing 1% HCl, at room temperature using a BV1000 vortex mixer (Benchmark Scientific Inc., New Jersey, NJ, USA). The mixture was centrifuged at 10,000× *g* for 15 min (Model Hermle Z326k, Hermle Labortechnik GmbH, Wehingen, Germany); the supernatant 2 mL was used for the determination of total phenolic content. Briefly, 9 µL of extract (supernatant) was mixed with 109 µL of Folin-Ciocalteu reagent, followed by 180 µL of (7.5% *w*/*v*) Na_2_CO_3_. The solution was then mixed and incubated for 5 min at 50 °C, and after cooling to 25 °C, the absorbance was measured at 760 nm (BMG LABTECH SPECTROstar Nano microplate reader, Ortenberg, Germany). Total phenolic compounds were calculated using a standard curve of gallic acid and expressed as mg of gallic acid equivalents per 100 g fruit leather.

### 2.4. Antioxidant Scavenging Capacity Using Ferric Reducing Antioxidant Power (FRAP) Assay

Total antioxidant scavenging activity of fruit leathers was determined using ferric reducing ability (FRAP), according to a method described by Ndou et al. [1] and the authors of [18]. The analysis was conducted using aqueous stock solutions containing 0.1 mol/L acetate buffer (pH 3.6), 10 mmol/L TPTZ [2,4,6-tris(2-pyridyl)-1,3,5-triazine] acidified with concentrated hydrochloric acid, and 20 mmol/L ferric chloride. Thereafter, the stocks were combined (10:1:1, *v*/*v*/*v*, respectively) in order to form the FRAP reagent just prior to analysis. Fruit leather samples (0.2 g) were briefly snap-frozen and homogenised in 2 mL of acetate buffer. Thereafter, ferric reducing ability was quantified using 15 μL aliquot of fruit extract, mixed with 220 μL of FRAP reagent solution. The absorbance was determined at 593 nm using a spectrophotometer (BMG LABTECH GmbH, Spectro- Star Nano, Ortenberg, Germany). The antioxidant activity (FRAP assay) was expressed as ascorbic acid equivalent antioxidant activity.

### 2.5. Quantification of Different Phenolic Compounds

UPLC coupled to a quadrupole time-of-flight (QTOF) mass spectrometer (UPLCQTOF/MS) was used to detect and quantify various phenolic compounds, as previously described for Natal plum by Ndou et al. [1] (2019). Freeze-dried samples (50 mg per replicate) were mixed in 1 mL of ethanol/water solution (70:30, *v*/*v*), ultrasonicated for 30 min, then centrifuged (Hermle Z326k, Hermle Labortechnik GmbH, Wehingen, Germany) at 1000 g at 4 °C for 20 min. Using a 0.22-μm polytetrafluorethylene filter, the supernatants were subsequently decanted and filtered. The resulting filtrate was analysed using a Waters Acquity™ UPLC coupled to a SYNAPT G2 QTOFMS (Waters, Milford, MA, USA) using Masslynx (V4.1) software. The type of column used for separation, the mobile phase used, gradient concentrations, and the running time set and flow rate were similar to the methodology described by Ndou et al. [1]. The UPLC was connected directly to a QToF-MS equipped with a source of electrospray ions operating in negative ESI mode with 15 V cone voltage, 275 °C desolvation temperature, and 650 L/h desolvation gas flow. For best resolution and sensitivity, the rest of the MS settings was optimised. Data were obtained both in resolution mode and in MSE mode by scanning from *m*/*z* 150 to 1500 m/z. Two channels of MS data were obtained in MSE mode, one using low collision power (4 V) and the other using a collision power accelerator (40–100 V) to acquire fragmentation information. For accurate mass determination, leucine enkephaline was used as a reference mass and the tool was calibrated using sodium format. Progenesis QI (NonLinear Dynamics, NC, USA) was used to identify the compounds. Standards purchased from four distinct cocktails were produced at each stage to help identify isomers and compounds with comparable elemental formulas [18]. Since calibration standards are not available for all the compounds identified, the compounds were semi-quantitatively measured against calibration curves set up using chlorogenic acid, catechin, luteolin, epicatechin, and rutin (Four cocktails, Sigma-Aldrich (Johannesburg, South Africa). Methanol (50%) in water containing formic acid (1%) was used for the cocktail preparation. HyStar 3.2 and Data Analysis 4.2 (Bruker Daltonics) software were used to analyse and calculate information [1]. A TargetLynx processing method (part of MassLynx) was adopted to quantify the compounds responsible for the main peaks using the cocktail standards in each chromatogram, as described by Ndou et al. [1].

The targeted compounds 4-hydroxybenzoic acid, protocatechuic acid, gallic acid, caftaric *acid*, dicaffeoyltartaric acid, chlorogenic acid, caffeic acid, caffeic acid, apigenin, catechin, epicatechin, kaempferol, and luteolin were quantified using HPLC, with a photo diode array ultraviolet detector, Model Flexar ^TM^ 89173-556 (PerkinElmer, Waltham, MA, USA), as described previously by Mpai et al. [18]. The column conditions and the mobile phase, flow rate, and gradient elution programme were according to Mpai et al. [18]. The chromatogram was read at 272, 280, 310, and 320 nm. The phenolic acids and flavonols were identified and quantified using pure external standards with 95% purity purchased from Sigma-Aldrich (Johannesburg, South Africa).

### 2.6. Proximate Analysis

Proximate analysis was performed on a selected fruit leather using AOAC standards methods [19].

### 2.7. Statistical Analysis

All experiments were carried out in triplicate. A complete randomised design was adopted; the data were analysed using analysis of variance (ANOVA) and means were compared using Fischer’s Least Significant Differences Test at *p* values 0.01 and 0.001, using the Genstat for Windows 204 13th Edition (2010) (VSN International, Hempstead, UK).

## 3. Results and Discussion

### 3.1. Colour Properties of Mango-Natal Plum Fruit Leathers

Colour properties of the five different kinds of fruit leathers M1, N1, MN5, MN3, and MN2 are shown in Table 1. Fruit leather containing only mango (M1) demonstrated the highest brightness (lightness *L**), whilst fruit leather made of Natal plum (N1) was the darkest due to the lower luminosity (*L**); Table 1. Addition of Natal plum to reduced quantities of mango pulp in MN3 and MN2 significantly affected brightness (*L**) of the fruit leathers (Table 1). Similarly, adding maqui berry extract (*Aristotelia chilensis* Mol. Stuntz) to apple and quince formulations made the fruit leathers darker (brown) [13] due to its higher anthocyanin content [20].

The colour coordinate positive *a** value relates to the red colour [10] (Ndou et al., 2019) of the fruit leather, which increased significantly when the Natal plum was added to the different ratios (5, 3 or 2) of mango fruit pulp (Table 1). A similar increase in *a** colour coordinates was noted when red acid calyx of roselle (*Hibiscus sabdariffa* L), rich in anthocyanin [21], was added to pineapple fruit leather [22]. In this study, although the redness in Natal plum dominates the yellow colour of the mango in the mango-Natal plum fruit leather, increasing Natal plum content did not significantly increase the redness of the mango-based fruit leather (Table 1). The yellowness (positive *b** values) of the mango pulp [23] in M1 and carotenoids are responsible for the yellowness in mango fruit leather [24]. Therefore, the addition of Natal plum to a mango pulp decreased the yellowness of the mango fruit leather by affecting the *L**, *a** and *b* * colour coordinates (Table 1). Natal plum, at the mature stage (red color), contains cyanidin derivatives [1], which could be reasonable for the observed changes in colour properties of the mango-Natal plum leathers. It is also evident from the ∆E (the change in colour difference) with reference to mango fruit leather (M1) as standard, Natal plum (N1) showed the highest colour difference due to its pinkish red colour followed by the Natal plum-mango (MN2) fruit leather, which contained the highest amount of Natal plum (Table 1).

### 3.2. Total Phenols and Antioxidant Property of Mango-Natal Plum Fruit Leathers

Total phenols were approximately seven times higher in Natal plum leathers (N1) than in mango leathers (M1) (Figure 2). Addition of Natal plum at different ratios increased the total phenols in the Natal plum-mango fruit leathers MN5, MN3, and MN2 (Figure 2). Although the increasing concentrations of Natal plum in the mango fruit leather did not significantly increase the total phenol content in MN5, MN3, and MN2, in general, adding Natal plum to mango pulp improved the total phenols content of mango leathers by approximately 25–27% (Figure 2). The ferric-reducing antioxidant power (FRAP) of the Natal plum-mango fruit leathers MN5, MN3, and MN2 showed a similar trend as the total phenol (Figure 3). Previous research has showed that low FRAP activity relates to the low phenolic content in the fruit [25]. In this study, adding Natal plum to mango fruit pulp significantly increased the antioxidant activity by 33.3–41.6% (Figure 3). Similarly, Torres et al. [14] demonstrated the improvement of total phenolic content and antioxidant activity by 40% and 45%, respectively, in a functional snack developed by fortification of apple leather with maqui berry extract. Phenolics and carotenoids present in mango pulp were responsible for the antioxidant activity of mango pulp [26]. On the contrary, adding red acid calyx of roselle (*Hibiscus sabdariffa* L) to pineapple minimised the total phenol content and antioxidant property of roselle-pineapple fruit leather, and the higher heating process (100 °C) led to the decarboxylation of phenolic acids [8,22]. However, the temperature used in this study, which was less than 100 °C, could have prevented the decarboxylation of phenolic acids in Natal plum.

Previous research showed that Natal plum contains a wide range of phenolic compounds [1], whilst no cyanidin derivatives were previously observed in mango pulp [27]; this might be the reason for the higher total phenol content of Natal plum leather compared to mango leather.

### 3.3. Cyanidin Derivatives in Mango-Natal Plum Fruit Leathers

The cyanidin derivative in the Natal plum is responsible for its red colour and is present in many berries [28]. Natal plum (N1) fruit leathers contained all three cyanidin derivatives, cyaniding-3-O-glucoside, cyanidin-3-O-β-sambubioside, and cyanidin-3-O-pyranoside (Table 2). These three cyanidin derivatives were reported in Natal plum at the red stage [1]; however, cyanidin-3-O-pyranoside was not detected in the mango-Natal plum leathers (MN5, MN3, MN2), which implied that the amount of Natal plum added did not significantly increase its concentration (*p* < 0.01) (Table 2). Although the concentrations of cyanidin-3-O-β-sambubioside and cyanidin-3-O-glucoside were significantly highest in Natal plum (N1), adding higher amounts of Natal plum (MN2) helped to retain these two compounds at higher levels (Table 2). The pH of the food matrix and temperature were reported to affect the stability of the cyanidin compounds [29]. Although the variability in the stability of the cyanidin compounds are dependent on the pH of the food matrix and the temperature, cyanidin-3-O-glucoside molecules were reported to be more stable than the cyanidin-3-O-pyranoside molecules [29]; thermal stability of cyanidin-3-O-β-sambubioside was also reported. However, formation of the quinonoid base and hemiketal (colourless) from the flavylium ion are favoured at higher temperatures [30]. Therefore, adding Natal plum, a rich source of cyanidin derivatives, to mango leathers, improved the functional compounds of mango fruit leather (Table 2). The drying temperature of the oven also affects the concentration of cyanidin derivatives; drying sour cherries at 50 °C reduced cyanidin-3-glucoside, compared to fresh ones [31]. In addition, extraction of the phenolic compounds affected the reduction in particle size, solvent types used for extraction, and the porosity of the dried samples [32].

Among the major anthocyanins, the bio availability of cyanidin-3-glucoside has been well documented [33]. Cyanidin-3-O-glucoside has been shown to exert insulin-like effects [34], anti-obesity effects [2], and anti-carcinogenic effects [35]; however, it has been shown that anthocyanins are bio-transformed in the gut before absorption. Many studies revealed that protocatechuic acid is one of the most likely major metabolites of anthocyanins [36,37]. It was demonstrated by Ormazabal et al. [38] that protocatechuic acid has the ability to regulate insulin responsiveness and inflammation during obesity. Findings of Ormazabal et al. [38] confirmed the advantageous effects of improving anthocyanin content in diets to combat against inflammation and insulin resistance in obesity.

### 3.4. Correlation between the Predominant Cyanidin Components of Colour Properties and the Antioxidant Activity

Cyanidin-3-O-beta sambubioside (R^2^ = 0.89) and cyanidin-3-O-glucoside chloride (R^2^ = 0.92) in the Natal plum-mango fruit leathers strongly contribute to total phenolic content and also positively correlate to antioxidant properties (FRAP assay) (Table 3); this supports the observed higher antioxidant property in mango leather enriched with Natal plum, especially M2 and M3, compared to the mango fruit leather (M1) shown in Figure 3 This observation confirmed that the inclusion of Natal plum to mango-based leather improved antioxidant activity. Both cyanidin components negatively correlated to the brightness (*L**) (R^2^ = −0.83, −0.81); similarly, the total phenols and antioxidant property correlated strongly and negatively with brightness (*L**) (R^2^ = 0.96) (Table 3). In addition, the colour coordinate *b**, which that relates to the yellowness of the fruit leather, strongly and negatively correlated to the antioxidant activity (R^2^ = −0.78) and total phenols (R^2^ = 0.97). Therefore, the darker Natal plum-mango fruit leathers are rich in antioxidant properties and can be regarded as a functional snack.

### 3.5. Flavonoids and Phenolic Acids in Mango-Natal Plum Fruit Leathers

Six dietary flavonoid aglycones and glycosides were quantified in the five types of leathers (N1, M1, MN2, MN3, MN5) namely, apigenin, catechin, epicatechin, kaempferol, luteolin, quercetin-3-O-rhamnosyl glucoside (rutin), quercetin-3-O-rhamnosyl galactoside, eriodictyol-7-O-glucoside, and naringenin-4-O-glucoside (Figure 4). Natal plum leather showed the highest concentration of epicatechin, rutin, quercetin-3-O-rhamnosyl galactoside, catechin, luteolin, naringenin-4-O-glucoside, and eriodictyol-7-O-glucoside (Figure 4). In mango fruit leather, the flavonoid concentrations were significantly lower and rutin, eriodictyol-7-O-glucoside, and quercetin-3-O-rhamnosyl galactoside, were not detected (Figure 4). Lower concentrations of flavonoids were detected in mango pulp [39]. However, the pulp of mango variety Tommy Aitkin, obtained from Ecuador, showed an absence of flavonoids [40]. The reduction in flavonoid concertation, or its disappearance, was associated with ripening [39]. Adding Natal plum to mango pulp in MN2 increased the concentration of epicatechin similar to the levels noted in Natal plum. Fruit leather MN2 retained the highest concentration of apigenin compared to the other fruit leathers, and the concentration of catechin content in fruit leather MN3 was similar to the levels noted in Natal plum. In general, fruit leathers MN2 showed higher levels of flavonoids than MN3. Furthermore, epicatechin, catechins, kaempferol, apigenin, rutin, and eriodictyol were reported to demonstrate many health benefits [41].

The quantified phenolic acids were quinic acid and chlorogenic acid. Quinic acid was high in N1, and M1 had the least (Table 4). Natal plum was previously reported to contain less than 1 mg/kg chlorogenic acid in the red fruit [1]. The high amount of this phenolic in Natal plum leather may imply that phenolics were concentrated during dehydration. As the proportion of Natal plum to mango in the leather increased, the amount of quinic and chlorogenic acid in the leather increased (*p <* 0.01). Chlorogenic acid was not detected in mango fruit leather or reported in fresh mango pulp [40]. Quinic acid is also a major bioactive chemical found in Saskatoon berry genotypes [42]. Thus, the presence of these phenolic acids increases the potential of mango-Natal plum leather as a functional snack. Total ion chromatograms of metabolites of Natal plum–mango (MN2) fruit leather, in comparison to Natal plum fruit leather in ESI-mode by UPLC–QTOF/MS are shown in Appendix A.

### 3.6. Proximate Analysis of Mango-Natal Plum Fruit Leather (MN2)

Since the mango-Natal plum, fruit leather MN2 showed the highest concentration of functional compounds and antioxidant activity, it was further analysed for proximate composition. The proximate composition analysis included moisture, dry matter, sugars, total fat, carbohydrate, protein, total ash, energy, total dietary fibre, and sodium (Table 5). Moisture content in MN2 was observed to be quite high. Generally, a moisture content of 15–25% is preferred for fruit leathers; however, it is affected by sugar content, acidy of the fruit, drying process, temperature, and humidity [5,9]. Azeredo et al. [9] demonstrated six months’ shelf life without preservatives for mango fruit leathers with low water activity (0.62), low pH (3.8), and a moisture content of 17.2%. Furthermore, although low moisture content can extend shelf life by preventing the fruit leathers from inhibiting microbial growth, the texture of the fruit leathers can be negatively affected due to extensive crispiness of the product [43,44]. The sugar content was 40.85 g/100 g—about 6.12 g of sugar for three servings of one fruit bar (5 g). According to the American Heart Association (AHA) [45], added sugar intake must be limited to less than 25 g sugar or 100 cal intake per day for children ranging from 2 to 18 years of age. Similarly, for men and women, 37.5 g (9 teaspoons) and 25 g of sugar intake per day, respectively, is adequate [45]. Therefore, three servings of one fruit bar (5 g) will contribute approximately towards ¼ portion of the 25 g that is allowable for children and women, and 1/6 portion of 37.5 g for men. In addition, this does not fall under the refined or processed sugar and is natural sugar found in the fruit. Refined sugar, which is an important contributor to dietary energy, has recently been a target of campaigns to reduce refined sugar intake [46]. Furthermore, better access to information on the amounts of sugar added to processed food is essential for appropriate monitoring of this important energy source [46]. The natural sugar present in Natal plum–mango fruit leather is fructose, with a number of micronutrients plus fiber. Natal plum–mango fruit leather contains 67.98 g/100g fiber. It is evident from previous research conducted by French and Read [47] that dietary fiber produces a viscous gel-like environment in the small intestine and this slows down gastric emptying, thereby facilitating a gradual release of sugar into the bloodstream. This avoids a glucose spike in the blood. Subsequently all the above-mentioned events lead to a reduced hunger sensation and eventually decrease food consumption. Digestion-resistant cellulose, hemicellulose, pectic substances, gums, mucilage, and lignin in dietary fibre [48] can be responsible for creating the viscous gel-like environment and have been proposed as important for a healthy gut microbiome [49]. Furthermore, the higher fibre content in this fruit leather may have also contributed to the high moisture content, since fibre is hydrophilic. Dietary fibre has been reported to lower the glycaemic indices of food. Supplementation of mango powder (10g) with carbohydrate, protein, fat, and fiber contents of 89.6%, 4.01%, 1.62%, and 13.4%, respectively, in obese individuals actually improved fasting glucose levels [50]. Thus, on this note, the sugar content in Natal-plum fruit leather snack may not offset any potential benefits of bioactive components.

In the presence of hydrophilic groups (such as those from pectin, fibre, and other soluble solids), drying can take longer, but longer drying times are associated with degradation of phytochemicals [32]. In this study, a lower temperature and shorter drying time were chosen in order to have a softer leather, as well as to preserve the phytochemicals.

The fruit leather MN2 was high in carbohydrates and low in proteins and fats (Table 5). The carbohydrate content in fruit leather MN2 was quite low, compared to blueberry fruit leather (89.00 mg/100g FW) and banana-pineapple-apple leather (80.00 ± 0.10 to 84.77 ± 0.06%) [51]. A lower carbohydrate content is associated with a lower calorific content of food. One hundred grams of fruit leather contains 1129.18 kJ (269.7 cal) (Table 5); therefore, three servings of one fruit bar (5 g) contains approximately 40 cal. The mango–Natal plum leather was also found to have low sodium content (Table 5). Strawberry leathers have been reported to contain 121.47 mg/kg of sodium [52], and this low sodium content makes it acceptable for people with high blood pressure and kidney problems. Furthermore, higher values of ash content relate to the higher mineral composition [53]. Mango-Natal plum fruit leather MN2 contained slightly higher ash content than that reported in the mixed fruit leather, which contained 40% banana, 40% pineapple, and 20% apple (1.20%). The visual appearance of mango fruit leather enriched with Natal plum (MN2) compared to the mango is illustrated in Figure 5.

## 4. Conclusions

Mango leather had a lower antioxidant activity and phenolic content compared to Natal plum leather. Increasing amounts of Natal plum in mango–Natal plum leather (MN2) significantly improved cyanidin components, flavonoids, and antioxidant activity. Mango–Natal plum leather (MN2) also contained carbohydrates, was high in fibre, and low in sodium, and can therefore be regarded as a potential functional snack.

## Figures and Tables

**Figure 1 foods-09-00431-f001:**
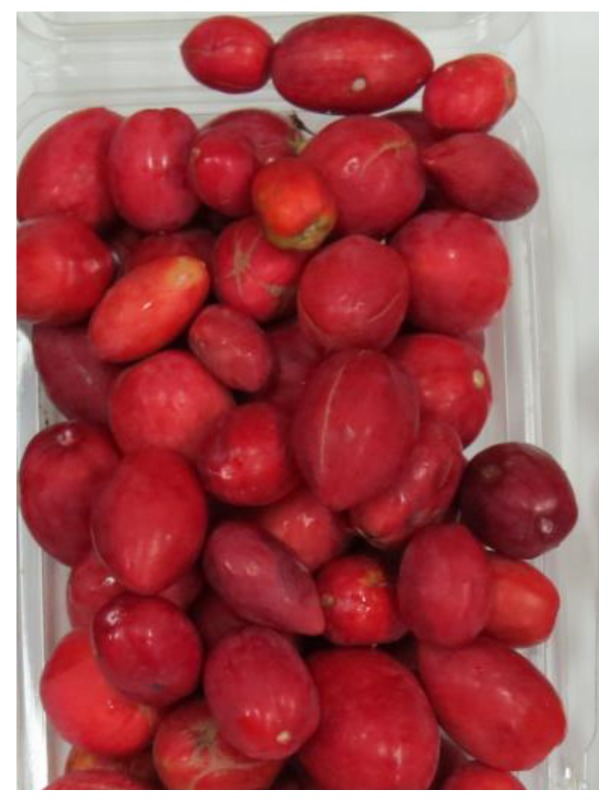
Natal plum (*Carissa macrocarpa*) at maturity stage 4 (red).

**Figure 2 foods-09-00431-f002:**
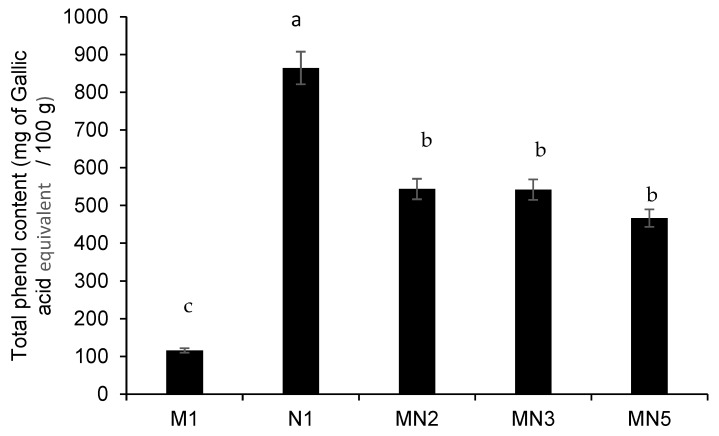
Total phenolic content of Natal plum-mango fruit leather at different ratios. Bars with different alphabets are significantly different *p <* 0.001 by Fisher’s protected least significant test.

**Figure 3 foods-09-00431-f003:**
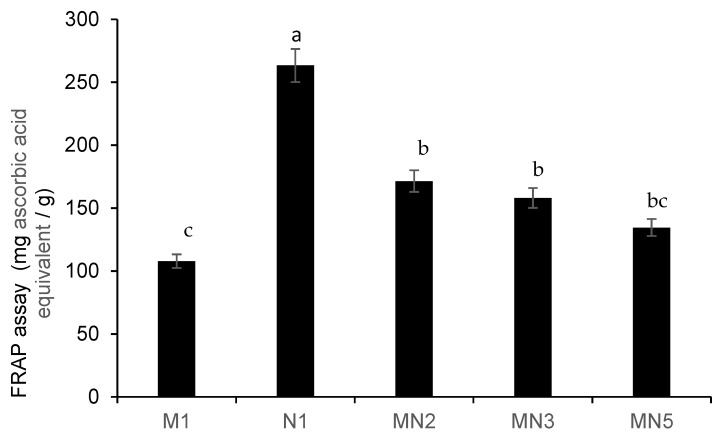
Antioxidant capacity (FRAP assay) of Natal plum-mango fruit leather at different ratios. Bars with different alphabets are significantly different *p <* 0.001 by Fisher’s protected least significant test.

**Figure 4 foods-09-00431-f004:**
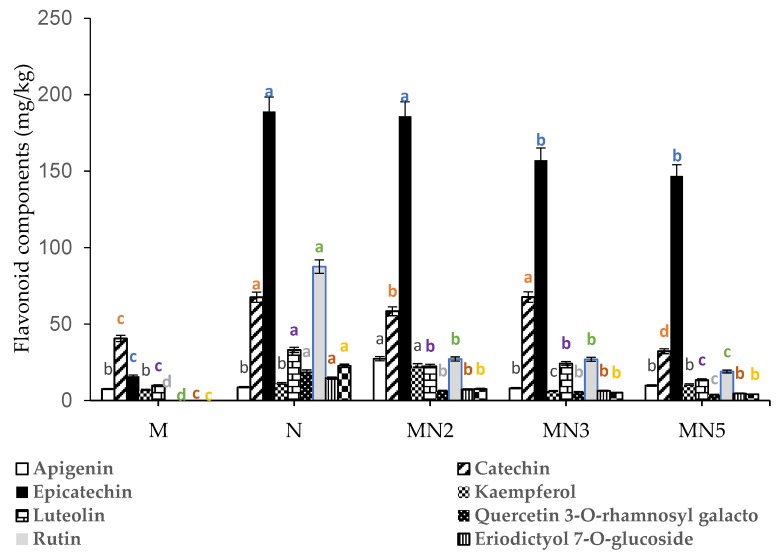
Flavonoid components of Natal plum-mango fruit leather at different ratios. Bars with different alphabets are significantly different *p <* 0.001 by Fisher’s protected least significant test. In addition, bars with specific coloured alphabets relate to specific flavonoid compounds.

**Figure 5 foods-09-00431-f005:**
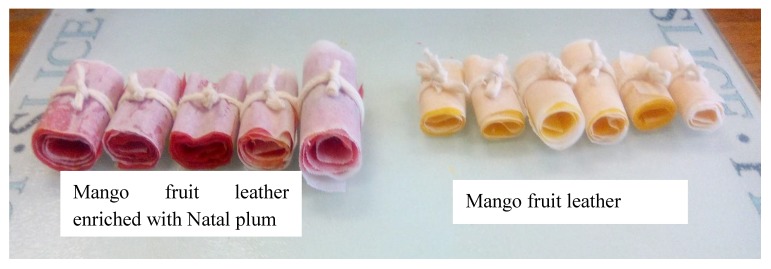
Mango fruit leather enriched with Natal plum.

**Table 1 foods-09-00431-t001:** Colour properties of Natal plum-mango fruit leather at different ratios.

	Colour Properties			
Fruit Leather Formulation	*L**	*a**	*b**	Δ*E*
M1 (Mango)	57.2 ± 1.5 *^a^	5.8 ± 0.5 ^c^	39.8 ± 1.7 ^a^	Standard
N1(Natal plum)	36.4 ± 0.4 ^d^	21.3 ± 0.9 ^b^	7.7 ± 0.4 ^e^	41.3 ^a^
MN2 (2M:1N)	41.6 ± 0.4 ^c^	27.7 ± 0.6 ^a^	16.2 ± 0.7 ^d^	35.8 ^b^
MN3(3 M:1N)	43.5 ± 0.6 ^c^	26.9 ± 0.6 ^a^	19.4 ± 0.7 ^c^	32.4 ^c^
MN5(5 M:1N)	46.1 ± 0.5 ^b^	27.4 ± 0.8 ^a^	22.9 ± 0.8 ^b^	29.6 ^d^

Total color difference (ΔE). Values (Means ± S.D*) with different superscripts in a column are significantly different *p* < 0.001 by Fisher’s protected least significant test. * standard deviation.

**Table 2 foods-09-00431-t002:** Concentration of cyanidin derivatives in Natal plum-mango fruit leather at different ratios.

Fruit Leather Formulation	Cyanidin-3-O-pyranoside (mg/kg)	Cyanidin-3-O-β_sambubioside (mg/kg)	Cyanidin-3-O-glucoside (mg/kg)
N1(Natal plum)	1.6 ± 0.01 a	46.9 ± 0.02 a*	52.5 ± 0.02 a
M1(Mango)	ND	8.6 ± 0.01 e	1.5 ± 0.04 e
MN2(2M:1N)	ND	29.2 ± 0.03 b	21.9 ± 0.01 b
MN3(3M:1N)	ND	20.9 ± 0.01 c	19.4 ± 0.02 c
MN5(5M:1N)	ND	18.9 ± 0.03 d	17.7 ± 0.01 d

Values (Means ± S.D) with different alphabets in a column are significantly different *p <* 0.01 by Fisher’s protected least significant test. * Standard deviation. ND = not detected.

**Table 3 foods-09-00431-t003:** Pearson’s correlation coefficients between the predominant cyanidin components of color properties and the antioxidant activity of Natal plum-mango fruit leather.

	Total Phenols	FRAP	Cyanidin-3-O-Beta Sambubioside	Cyanidin-3-O-Glucoside Chloride
	Correlation coefficients (R^2^)			
Total phenols		0.869	0.90	0.92
FRAP			0.97	0.97
*L**(C)	−0.96	−0.76	−0.83	−0.81
*a**(C)	0.39	0.1	0.17	0.15
*b**(C)	0.97	−0.78	−0.85	−0.82

**Table 4 foods-09-00431-t004:** Concentration of quinic and chlorogenic acids in the Natal plum-mango fruit leathers at different ratios.

Fruit Leather Formulation	Quinic Acid (mg/kg)	Chlorogenic Acid (mg/kg)
N1	1678.1 ± 1.30 ^a^*	16.6 ± 0.03 ^a^
M1	188.6 ± 0.6 ^e^	Nd
MN5	565.8 ± 0.12 ^d^	3.2 ± 1.00 ^c^
MN3	646.7 ± 0.50 ^c^	3.5 ± 0.06 ^c^
MN2	708.0 ± 0.10 ^b^	5.4 ± 0.09 ^b^

Values (Means ± S.D) with different superscripts in a column are significantly different *p <* 0.001 by Fisher’s protected least significant test. * Standard deviation, Nd—not detected.

**Table 5 foods-09-00431-t005:** Proximate composition of Natal plum-mango fruit leather (MN2).

Proximate Composition	Results g/100 g
Moisture	32.02 ± 0.34
Dry matter	67.98 ± 0.82
Sugars sum	40.85 ± 2.86
Total fat	0.36 ± 0.03
Carbohydrates	63.51 ± 0.12
Protein	1.63 ± 0.08
Total ash	1.42 ± 0.07
Energy	1129.18 kJ
Total Dietary Fibre (TDF)	7.29 ± 0.87
Sodium (Na)	0.005 ± 0.01

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
