# Peer review of "Enrichment of Mango Fruit Leathers with Natal Plum (Carissa macrocarpa) Improves Their Phytochemical Content and Antioxidant Properties"

_foods, 2020, doi:10.3390/foods9040431_

Round 1
Reviewer 1 Report
Reviewer Evaluation:
This research paper assesses the addition of mango fruit leathers with an indigenous fruit, Natal plum. Overall, this paper is quite interesting, but I question the health benefit of this particular functional food as it is high in sugar, which might offset any potential benefit of the bioactive components. Additionally, there is issues with the statistics and results layout that need a little attention.
General Comments:
The title requires slight grammar edits, I suggest the following title “Enrichment of Mango Fruit Leathers with Natal Plum (Carissa macrocarpa) Improves their Phytochemical Content and Antioxidant Properties” In vitro (no dash), change throughout Line 137, why were the samples snap frozen before the experiment? Should they not have just been thawed? Missing lines for navigating through the manuscript from page 10 onwards. It would have been good to have an image of all the different coloured fruit leathers side by side as it seems in figure 5 seems to be only 2 types, correct? (They look tasty though). Include author contributions. A representative chromatogram may enhance the results section of the manuscript, particularly since you did targeted analysis. This is decent analysis to conduct and should be prominent in the manuscript. The discussion section could benefit from a mention of study limitations.
Major Points of improvement:
You are claiming this product to be a healthy functional fruit snack and you have made good points from lines 60-66. However, how much is a typical portion size? The sugar content of the snack is very high and one must question the validity of eating the snack when you could just eat the fruit and get all the benefits of the antioxidants, phytochemicals, and fibre of eating the whole fruit. Do fruit leathers have a negative impact on glucose levels in the blood? Will the high sugar levels offset and possible benefits of cyanidin-3-glucoside. Although it is clear that the fruit does economically help indigenous populations, which is a very favourable outcome. Line 153, you say your methodology was similar to the one referenced in text. Can you highlight what if there were changes. Figure 2 – how is there not statistical difference between M1 and MN5, when there is a difference between N1 and MN2. Fisher’s LSD is quite forgiving analysis so this doesn’t seem right. Figure 3 as above. Are you sure these stats are correct. Why was proximate analysis of only MN2 conducted? It stands to reason that different compositions may lead to different analysis.
Minor Points:
Line 18 – …of indigenous what? …fruit?
Lines 171-173 – no need for italics, why is the font different
Line 185 – p value (small letter p in italics)
Table 1 – caption should be above the table. Same comment above in relation to the p value. Underneath the table include the abbreviations of L*, a*, b* etc. Also state the statistical test used.
Line 242 and Figure 4 – letters not alphabets.
Figure 3 – align your letters of statistical significance directly above the bars.
Line 263 – 266 font size and style issues
Line 267 – bioavailability
Lie 267 – cyanidin-3-glucoside
Table 2 – you say superscripts, but you haven’t input superscripts. E.g. xa
Section 3.6 – approx. 20-22 lines there is spacing issues.
Table 5 – there is no need for square brackets around the SD
Concluding Remarks:
Overall, the research seems interesting but requires further review by the authors. There are some presentation mistakes made and there are questions to be addressed. In particular, the authors should address the issues surrounding the high sugar content. Is there evidence that the cyanidin-3-glucoside modulates blood sugar levels despite the high sugar content. Or is this issue insignificant due to portion size. In which case, is an effective dose delivered in a normal portion size to potentially benefit a consumer. These issues could be discussed further.
Author Response
AnswerOf course, the fruit bar has sugar. For example, mango fruit has sugar but the novelty is the additional of anthocyanin compounds. In Biomol Ther
1.This research paper assesses the addition of mango fruit leathers with an indigenous fruit, Natal plum. Overall, this paper is quite interesting, but I question the health benefit of this particular functional food as it is high in sugar, which might offset any potential benefit of the bioactive components.
Answer
Of course, the fruit bar has sugar. For example, mango fruit has sugar but the novelty is the additional of anthocyanin compounds. In Biomol Ther (Seoul) 2013; 21(4): 284-289 , the folowing study showed, that the
‘’Antidiabetic and Beta Cell-Protection Activities of Purple Corn AnthocyaninsPurple corn anthocyanins also reported to contain insulin secretion activity (Jayaprakasam et al., 2005). Cyanidine 3-glucoside-rich purple corn anthocyanins improved the high fat diet-induced obesity and hyperglycemia in mice (Tsuda et al., 2003). Cyanidine 3-glucoside reported to ameliorate hyperglycemia and insulin sensitivity due to downregulation of retinol binding protein 4 expression in diabetic mouse (Sasaki et al., 2007) on this note we still argue that this product has functional properties but we have to do in vitro antidiabetic enzyme activities and cellular uptake.
2.Additionally, there is issues with the statistics and results layout that need a little attention.
answer -It has been checked and revised
3. General Comments:
The title requires slight grammar edits, I suggest the following title “Enrichment of Mango Fruit Leathers with Natal Plum (Carissa macrocarpa) Improves their Phytochemical Content and Antioxidant Properties”
answer – revised as suggested
4. In vitro (no dash) – revised as suggested
, change throughout Line 137, why were the samples snap frozen before the experiment? It was mentioned in the original text in line 105 to 106 that ‘’A set of 10 replicates of each fruit leather were snap-frozen in liquid nitrogen for antioxidant and phytochemical analysis’’
Answer- Snap freezing is done to stop the changes in biochemical reaction leading the changes in phenolic compounds.
5. Should they not have just been thawed?.
Answer-The analysis was conducted at 10 C in a cold room.
Missing lines for navigating through the manuscript from page 10 onwards.- Answer -Sorry, do not understand the comment. Page lines are included
It would have been good to have an image of all the different coloured fruit leathers side by side as it seems in figure 5 seems to be only 2 types, correct? (They look tasty though).
Answer-Sorry we do not have photographs with higer resolution
6. Include author contributions. A representative chromatogram may enhance the results section of the manuscript, particularly since you did targeted analysis. This is decent analysis to conduct and should be prominent in the manuscript. The discussion section could benefit from a mention of study limitations. –
Answer-The chromatograms are given in supplementary files. Kindly note we included the chromatograms of Natal plum fruit leather and the MN2 Natal plum and mango fruit leather. Other chromatograms are not included. We will not include all the chromatograms.
7. Major Points of improvement:
You are claiming this product to be a healthy functional fruit snack and you have made good points from lines 60-66.
However, how much is a typical portion size? The sugar content of the snack is very high and one must question the validity of eating the snack when you could just eat the fruit and get all the benefits of the antioxidants, phytochemicals, and fibre of eating the whole fruit.
Do fruit leathers have a negative impact on glucose levels in the blood?
Answer- According to the American Heart Association (AHA) https://www.healthline.com/nutrition/how-much-sugar-per-day#section, The sugar content was 40.85 g/100 g and it is about 6.12 g of sugar for three servings of one fruit bar (5 g). According to the American Heart Association (AHA), recommends less than 25 grams (6 spoons) of added sugar intake per day for children ranging from two and 18. Similarly, for men and women 37.5 g (9 teaspoons) and 25 g (6 teaspoons) of added sugar intake per day was recommended. Therefore, three servings of one fruit bar (5 g) will contribute approximately towards ¼ portion of 24.00 g that is recommended for the children and women and 1/6 portion of 37.5 g for men. In addition, this does not fall under the refined or processed sugar and it is natural sugar found in the fruit. Refined sugar is an important contributor to dietary energy, but recently it is a potential target for campaigns to reduce refined sugar intake (Somerset, S.M. 2003). Furthermore, better access to information on the amounts of sugar added to processed food is essential for appropriate monitoring of this important energy source. (Somerset, S.M. 2003). The natural sugar present in Natal plum –mango fruit leather is fructose, with a number of micronutrients plus fiber. Natal plum –mango fruit leather contains 67.98 g /100g fiber. It was evident from the previous research conducted by French and Read [1], that the dietary fiber produces a viscous gel-like environment in the small intestine, and result in inhibition of gastric emptying and contributing to the decrease in energy intake, slower absorption rate of sugar into the bloodstream. Subsequently all the above-mentioned events leads to reduce hunger sensation and eventually decrease food consumption. Thus, on this note Natal-plum fruit leather snack does not offset any potential benefit of the bioactive components.
Digestion-resistant cellulose, hemicellulose, pectic substances, gums, mucilage and lignin in the dietary fibre [46] can be responsible for creating the viscous gel-like environment and has been proposed to be important for a healthy gut microbiome [47]. Furthermore, the higher fibre content in this fruit leather may have also contributed to the high moisture content since fibre is hydrophilic. Dietary fibre has been reported to lower the glycaemic indices of food. Thus, on this note the sugar content in Natal-plum fruit leather snack does not offset any potential benefit of the bioactive components.
[45]. American Heart Association (AHA) recommended sugar intake. Available online:
https://www.healthline.com/nutrition/how-much-sugar-per-day#section. (Accessed 3rd of March 2020)
[46]. Somerset, S.M. Refined sugar intake in Australian children. Public Health Nutr.2003, 6, 809-813.
[47]. French, S.J.; Read, N.W. Effect of guar gum on hunger and satiety after meals of differing fat content: relationship with gastric emptying. Am J Clin Nutr. 1994, 59, 87-91
Will the high sugar levels offset and possible benefits of cyanidin-3-glucoside. Yes please see the abovementioned explanation. Although it is clear that the fruit does economically help indigenous populations, which is a very favourable outcome.
Line 153, you say your methodology was similar to the one referenced in text. Can you highlight what if there were changes?
Answer-There are no changes in the methodology adopted
Figure 2 – how is there not statistical difference between M1 and MN5, when there is a difference between N1 and MN2. Fisher’s LSD is quite forgiving analysis so this doesn’t seem right.
Answer-Sorry typo mistake corrected in the revised version
Figure 3 as above. Are you sure these stats are correct.
Answer- Yes correct
Why was proximate analysis of only MN2 conducted? It stands to reason that different compositions may lead to different analysis.
Answr-The reason was given in
Mentioned in the original version in 324 to 325 as ‘’Since the mango-Natal plum fruit leather MN2 showed the highest concentration of functional compounds and antioxidant activity, it was further analysed for proximate composition’’
Minor Points:
Line 18 – …of indigenous what? …fruit?
Ansewr-Sorry not clear
Lines 171-173 – no need for italics, why is the font different-revised and highlighted in blue fonts
Line 185 – p value (small letter p in italics)
Answer-revised and highlighted in blue fonts
Table 1 – caption should be above the table. Same comment above in relation to the p value. Underneath the table include the abbreviations of L*, a*, b* etc. Also state the statistical test used.
Answer-Revisions highlighted in blue fonts
Line 242 and Figure 4 – letters not alphabets.-You mean with bars but alphabets are present.
Answer-Yes those are vales of 0 but still the statistical donations must be given.
Figure 3 – align your letters of statistical significance directly above the bars. Answer-Revised accordingly
Line 263 – 266 font size and style issues
Answer-Revised accordingly. Probably got changed during formatting
Line 267 – bioavailability
Answr-sorry it has to be biotransformation
Line 267 – cyanidin-3-glucoside
Answer-Revised accordingly
Table 2 – you say superscripts, but you haven’t input superscripts. E.g. xa Answer-Revised accordingly
Section 3.6 – approx. 20-22 lines there is spacing issues.
Answer-Revised accordingly
Table 5 – there is no need for square brackets around the SD-
Answer-Revised accordingly
Concluding Remarks:
Overall, the research seems interesting but requires further review by the authors. There are some presentation mistakes made and there are questions to be addressed. In particular, the authors should address the issues surrounding the high sugar content. Is there evidence that the cyanidin-3-glucoside modulates blood sugar levels despite the high sugar content. The explanation has been given and addressed .
Or is this issue insignificant due to portion size. In which case, is an effective dose delivered in a normal portion size to potentially benefit a consumer. These issues could be discussed further.
Answer- Mentioned above
Reviewer 2 Report
This manuscript was objected to investigate chemical composition and antioxidant capacity for the fruit leather mixed with Natal plum and mango. The reserach topic seems to be interesting to food scientists, however, the research design and hypothesis seems to be little importance to this field. Synergistic or additional effect can be explored in biological activity, but, chemical composition may be no meaning when combined. This manuscript has little scientific originality.
Author Response
Comments and Suggestions for Authors
This manuscript was objected to investigate chemical composition and antioxidant capacity for the fruit leather mixed with Natal plum and mango. The reserach topic seems to be interesting to food scientists, however, the research design and hypothesis seems to be little importance to this field. Synergistic or additional effect can be explored in biological activity, but, chemical composition may be no meaning when combined. This manuscript has little scientific originality.
The reserach topic seems to be interesting to food scientists, however, the research design and hypothesis seems to be little importance to this field
- much appreciated the comment
however, the research design and hypothesis seems to be little importance to this field.-
- not sue which field of research reviewer is referring
Synergistic or additional effect can be explored in biological activity, but, chemical composition may be no meaning when combine
- We are currently busy with the biological activity but unfortunately, that part of the study cannot be included here.
This manuscript has little scientific originality.
- This must be the reviewer’s opinion but we appreciate it. However, the novelty is based on the enrichment of the product and improving the antioxidant activity.
The editor of this journal will make final decision. Many thanks again for your time.
Reviewer 3 Report
In the article, improved content of phenolic antioxidants and antioxidant activity by enrichment of mango fruit leathers with Natal plum has been provided as well as colour of the product.
To the article, I have next comments and recommendations:
- 45:…are another practical way…
- 64: …for health food…
- 104: 4x10-3 kg/m2/d…
- 118: Db = colour difference is calculated
- 136: Thereafter, the stocks…
- 151: …types of column used…
- 165, 167: instead cocktail use solution; solutions of four concentrations
- 171- 173: Why in italics?
- 225-229: Fig. 3.
- In the whole article attention must be paid to the correct writing of the names of anthocyanins, e.g. cyanidin-3-O-pyranoside; cyanidin-3-O-glucoside; etc. It should be corrected and checked in all manuscript and tables.
- In the text the same format of letters should be used, correct l. 263-266
- Correct like R2 = 0.89 and in all other cases.
- Figure 4:…columns with different letters…
- …acidity of the fruit…; the cyanidin components.
- Data in Table 5: In Materials and method: methods of the determination are not cited and referred or described.
- Author contributions are missing.
- In References correct the abbreviations of the journals names, e.g. Food Chem. You used some abbreviations with dots, but most are without them. They should be corrected and unified according to the journal rules.
- References [21] and [24], [45] are in bold. Correct the format.
- Check the References, e.g. [28] Seeram, N. P.; …
Author Response
Reviewer 3
Comments and Suggestions for Authors
In the article, improved content of phenolic antioxidants and antioxidant activity by enrichment of mango fruit leathers with Natal plum has been provided as well as colour of the product.
To the article, I have next comments and recommendations:
- 45:…are another practical way
- Answer- Revised according to the comment
- 64: …for health food…
- Answer-Revised according to the comment and highlighted in blue font
- 104: 4x10-3 kg/m2/d… Revised according to the comment and highlighted in blue font
- 118: Db = colour difference is calculated
- Answer-∆b* = Colour difference
- 136: Thereafter, the stocks
- Answer--… Revised according to the comment and highlighted in blue font
- 151: …types of column used-
- Answer-wrong it must be type of column…
- 165, 167: instead cocktail use solution; solutions of four concentrations- Answer-We prefer to keep as it is
- 171- 173: Why in italics? Revised
- 225-229: Fig. 3. Revised
- In the whole article attention must be paid to the correct writing of the names of anthocyanins, e.g. cyanidin-3-O-pyranoside; cyanidin-3-O-glucoside; etc. It should be corrected and checked in all manuscript and tables. Revised
- In the text the same format of letters should be used, correct l. 263-266- Revised
- Correct like R2 = 0.89 and in all other cases
- Answer-it must be written as R2 and based on positive or negative correction the –ve is given
Figure 4:…columns with different letters…
Answer-it is Figure 3. Flavonoid components of Natal plum-mango fruit leather at different ratios. Bars with different alphabets are significantly different p<0.001 by Fisher’s protected least significant test. In addition, bars with specific coloured alphabets relate to specific flavonoid compound.
- …acidity of the fruit…; the cyanidin components.
- Answer-We prefer to keep it as sugar content, acidy of the fruit,
- Data in Table 5: In Materials and method: methods of the determination are not cited and referred or described.
- Answer-Since it is a common method, Mentioned as proximate analysis was performed on a selected fruit leather using AOAC standards methods [19].
- Author contributions are missing.
- Answer-Included
- In References correct the abbreviations of the journals names, e.g. Food Chem. You used some abbreviations with dots, but most are without them. They should be corrected and unified according to the journal rules. Answer- Revised
- References [21] and [24], [45] are in bold. Correct the format. Revised
- Check the References, e.g. [28] Seeram, N. P.; … Revised
Round 2
Reviewer 1 Report
The authors have made changes to improve the manuscript.
Some points to consider. AHA don't 'recommend' levels of added sugar, that implies that adults or children should proactively get added sugar. It is only a small grammar issue, but this really should be paid attention to when others in your field are to read your study. Similarly, while the study shows some changes in the antioxidant properties etc. some of the intended claims or potential benefits of the product are based on the composition of the product, but these effects are unfounded without additional experiments. Examining your chromatograms, quinic acid doesn't seem to have a distinctive peak. How are you so sure its quinic acid.
Author Response
- Some points to consider. AHA don't 'recommend' levels of added sugar, that implies that adults or children should proactively get added sugar. It is only a small grammar issue, but this really should be paid attention to when others in your field are to read your study.
- An
AHA Sugar Recommendation
To keep all of this in perspective, it’s helpful to remember the American Heart Association’s recommendations for sugar intake.
- Men should consume no more than 9 teaspoons (36 grams or 150 calories) of added sugar per day.
- For women, the number is lower: 6 teaspoons (25 grams or 100 calories) per day. Consider that one 12-ounce can of soda contains 8 teaspoons (32 grams) of added sugar! There goes your whole day’s allotment in one slurp.
- Check https://www.heart.org/en/healthy-living/healthy-eating/eat-smart/sugar/how-much-sugar-is-too-much
- Based on the above mentioned statement, it is revised as followes in the text as ''According to the American Heart Association (AHA), [45] added sugar intake must be limited and it recommends less than 25 grams sugar or 100 cal intake per day for children ranging from two and 18. Similarly, for men and women 37.5 g (9 teaspoons) and 25 g sugar intake per day was recommended''.
4. Examining your chromatograms, quinic acid doesn't seem to have a distinctive peak. How are you so sure its quinic acid. Rectified
Reviewer 2 Report
The reviesed version might be seemed to be reasonable. Before publication, in the introduction, the importance in the perspective of scientific view should be added. The relation to antioxidant activity for inclusion of Napal plum to mango-based leather should be explained.
Author Response
- Introduction, the importance in the perspective of scientific view should be added
- Answer- Sufficient information and the loggical evidenct has been included in the introduction. -No changes
- The relation to antioxidant activity for inclusion of Napal plum to mango-based leather should be explained
- Answer- correlated to the antioxidant property (FRAP assay) (Table 3) and further supports the observed higher antioxidant property in mango leather enriched with Natal plum especially M2 and M3 comapaired to the mango fruit leather (M1) shown in Fig 2.